# Nanoarchitecture and dynamics of the mouse enteric glycocalyx examined by freeze-etching electron tomography and intravital microscopy

Willy W. Sun[1,2,5], Evan S. Krystofiak[1,5], Alejandra Leo-Macias[1], Runjia Cui[1], Antonio Sesso[3], Roberto Weigert [4], Seham Ebrahim[4] & Bechara Kachar [1*]

The glycocalyx is a highly hydrated, glycoprotein-rich coat shrouding many eukaryotic and prokaryotic cells. The intestinal epithelial glycocalyx, comprising glycosylated transmembrane mucins, is part of the primary host-microbe interface and is essential for nutrient absorption. Its disruption has been implicated in numerous gastrointestinal diseases. Yet, due to challenges in preserving and visualizing its native organization, glycocalyx structure-function relationships remain unclear. Here, we characterize the nanoarchitecture of the murine enteric glycocalyx using freeze-etching and electron tomography. Micrometer-long mucin filaments emerge from microvillar-tips and, through zigzagged lateral interactions form a three-dimensional columnar network with a 30 nm mesh. Filament-termini converge into globular structures ~30 nm apart that are liquid-crystalline packed within a single plane. Finally, we assess glycocalyx deformability and porosity using intravital microscopy. We argue that the columnar network architecture and the liquid-crystalline packing of the filament termini allow the glycocalyx to function as a deformable size-exclusion filter of luminal contents.

[1] Laboratory of Cell Structure and Dynamics, National Institute on Deafness and Other Communication Disorders, National Institutes of Health, Bethesda, MD 20892, USA. [2] Neuroscience and Cognitive Science Program, University of Maryland, College Park, MD 20740, USA. [3] Sector of Structural Biology, Institute of Tropical Medicine, University of São Paulo, Sao Paulo, SP 05403, Brazil. [4] Laboratory of Cellular and Molecular Biology, National Cancer Institute, National Institutes of Health, Bethesda, MD 20892, USA. [5] These authors contributed equally: Willy W. Sun, Evan S. Krystofiak *email: kacharb@nidcd.nih.gov

The intestinal epithelium is the largest interface between our bodies and the external environment[1]. In the small intestine, thousands of actin-based membrane protrusions called microvilli on the apical surface of each enterocyte serve as the site of nutrient absorption. At the same time, the gut luminal environment is unique in its inhabitation by a vast and diverse microbial population, including residential symbiotic microbiota, and potentially pathogenic microorganisms that accompany food intake[2,3]. The intestinal mucosa is also exposed to mechanical stresses associated with peristalsis and luminal content propulsion onward along the intestinal tract[4]. To protect the intestinal mucosa from pathogens and mechanical stresses, epithelial cells generate protective layers comprised primarily of secreted and transmembrane glycoproteins that line the entirety of the intestinal tract. In the small intestine, the glycocalyx layer directly covers the entire surface of the epithelial cells[5–7] while the overlaying lubricant mucus layer is thin and discontinuous[7,8]. The glycocalyx comprises highly diverse glycoproteins and glycolipids expressed on the epithelial cell membrane, many of which serve as receptors for bacterial adhesion[7,9–11]. The glycocalyx thus act as attachment sites for normal flora to limit colonization by pathogens, in addition to functioning as size-selective diffusion barrier to exclude deleterious bacteria and viruses[12–14]. In addition to these protective roles, the intestinal glycocalyx contributes to the lubrication and hydrophobicity of the mucosal surface[15], prevents mucosal auto-digestion and ulceration, participates in cellular signaling, and functions as selective diffusion barrier for both endogenous and exogenous substances. Given these multitude roles in intestinal function and homeostasis, it is not surprising that glycocalyx impairment is implicated in a number of diseases of the intestinal tract, including inflammatory bowel disease and cancer[16].

Transmembrane mucins MUC1, MUC3, MUC4, MUC12, MUC13, and MUC17 are expressed by the intestinal mucosal epithelial cells and are presumed to be the main components of the glycocalyx[10,17,18]. The extracellular domains of these mucins range from 500 to 5000 amino acids in length consisting largely of mucin tandem repeats[17,19]. These repeats are rich in amino acids proline, serine, and threonine with extensive O-glycosylation on the serine and threonine residues[20]. This heavy sugar modification introduces steric hindrance and a negative charge repulsion, causing the mucins to assume a rod-like conformation[20,21]. Strong glycosylation of mucin domains also establishes highly hydrophilic regions along the protein, which may affect structuring of local water molecules, mucin filament interactions, and the overall mechanical properties of the glycocalyx layer[22,23]. Indeed, changes in the level of mucin glycosylation have been implicated in various pathological conditions[24].

Despite the wealth of biochemical data on both gel-forming and transmembrane mucins, progress in obtaining molecular-level details of mucin organization has been limited[25,26]. The glycocalyx in its native state is intrinsically difficult to study because of its highly hydrated nature. Conventional electron microscopy requires removal or replacement of water; this dehydration causes the glycocalyx structure to denature, leading to collapse of the filament network. Previous reports on the structure of the intestinal glycocalyx described its structure as made of densely entangled filaments forming a fuzzy coat covering the brush border[27].

Freeze-etching is a replica-based cryogenic technique where frozen water is removed by etching or sublimation keeping the glycocalyx organization near its native state[28]. An early freeze-etching study of cryoprotected rat intestinal tissue with limited etching showed that the glycocalyx is made of parallel filaments extending toward the lumen[29]. In this study, we used deep-etching and electron tomography to reveal the detailed three-dimensional (3D) organization of the enteric glycocalyx approximating its native state. We complement this approach with intravital imaging of the glycocalyx within the intestinal lumen of live, anesthetized mice to directly explore the porosity and deformability of this multifunctional layer in vivo, thus providing the most comprehensive structural framework for the enteric glycocalyx to date.

## Results

**Glycocalyx consists of a columnar filament network.** The intestinal epithelium is the single-cell layer that lines the entire lumen of the gastrointestinal tract. In the small intestine, it is convoluted into villi, projections that serve to increase the absorptive mucosal surface area (Fig. 1a). The epithelial layer comprises predominantly columnar epithelial cells (enterocytes), whose apical surface is covered in microvilli (Fig. 1a), actin-based protrusions that further increase the absorptive surface tenfold[30] and are collectively known as the brush border. The glycocalyx forms a layer over the microvillar surface (Fig. 1a).

To first ensure the retention and continuity of the glycocalyx in our preparation, we examined cryostat sections of adult mouse small intestine labeled with fluorescently tagged wheat germ agglutinin (WGA), which can bind sialic acid and N-acetylglucosaminyl moieties[31] that are known to decorate the proline-, threonine-, and serine-rich domains found in mucins[32]. We observed WGA immunoreactivity directly above the brush border identified by phalloidin labeling of actin (Fig. 1b). We also labeled the sections with an antibody against the conserved SEA domain (Sperm protein, Enterokinase, and Agrin domain) of human MUC17. This antibody is expected to label mouse MUC3 (the structural homolog of human MUC17)[33] and is likely to bind other SEA domain-containing mucins[19]. The MUC17 immunoreactivity showed a similar labeling pattern as the WGA consisting of a continuous layer above the brush border (Fig. 1c).

We next examined freeze-etching replicas of the apical surface of mouse small intestine to visualize the ultrastructural details of the enteric glycocalyx (Fig. 1d–h). The deep-etching procedure involves a fixation step and rinsing of the tissue with distilled water to remove any soluble components (salt and small molecules) that would overlay the surface structures upon sublimation of the water. Low magnification of freeze-etch replicas shows that the glycocalyx on the brush border surface forms a continuous (Fig. 1d), transcellular layer with no boundaries between cells (Supplementary Fig. 1), consistent with the mucin staining. The layer had a uniform thickness of $1.0 \pm 0.1 \, \mu m$ ($n = 53$), which was comparable to the length of microvilli ($1.02 \pm 0.05 \, \mu m$, $n = 23$) and was composed of densely interwoven columnar filaments (Fig. 1e). Multiple glycocalyx filaments emerge specifically from the distal ends, or tips, of individual microvilli and almost immediately form lateral interactions, including with filaments emerging from adjacent microvilli (Fig. 1f and Supplementary Fig. 1a). While most filaments appear to coalesce to form the micron-long glycocalyx network and are likely made of the long mucins, we cannot exclude that some of the filament emerging from the microvilli tips correspond to shorter transmembrane mucins (Supplementary Fig. 1a). To verify that the columnar filaments and their lateral interactions are not artifacts of the fixation step prior to freezing, we also performed direct-freezing and deep-etching in unfixed tissue. In this case, while the filaments and surface structures are slightly obscured by the soluble components that remain after etching of the ice, the glycocalyx filaments and their lateral interactions (Fig. 1g) were comparable to those observed in the fixed tissues (Fig. 1f). Figure 1f also highlights the clear distinction between the inter-microvilli links[34] and the glycocalyx filaments that emerge

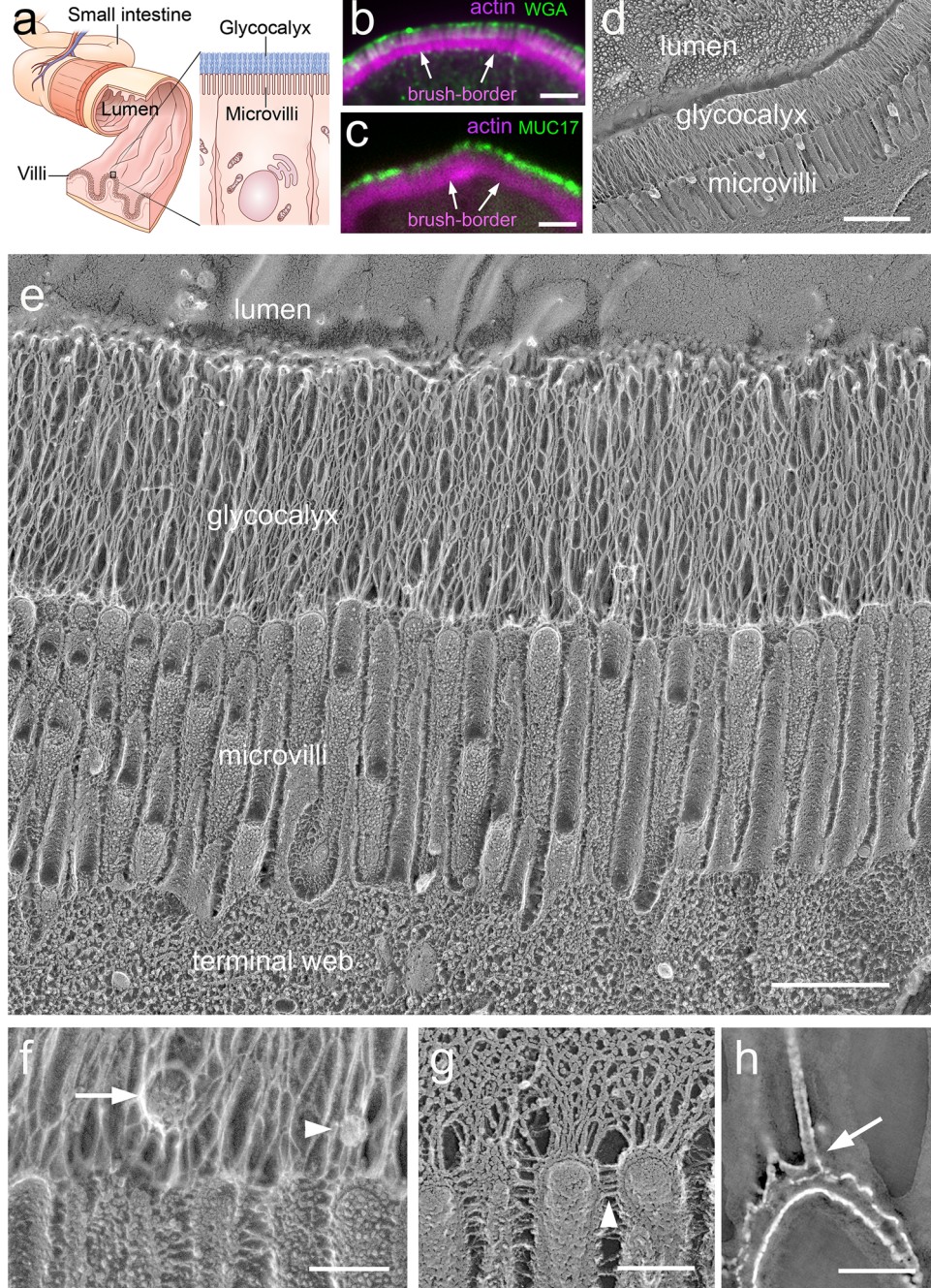

**Fig. 1 The enteric glycocalyx forms a uniform transcellular layer blanketing the microvilli-rich brush border. a** Schematic of lumen of the small intestine, which is convoluted into villi. Close-up view of an enterocyte showing the glycocalyx layer over its microvilli. **b, c** Cryo-sections of mouse small intestine were immunolabeled with (top) fluorescently tagged WGA (green) and (bottom) human anti-MUC17 (green) to highlight the glycocalyx layer over the actin-rich brush border (magenta). **d** Electron micrograph of a freeze-etch replica of mouse small intestine showing the stratified organization of the microvilli-rich brush border and glycocalyx layers. **e** Close-up view of the apical region of an enterocyte, showing glycocalyx separating the brush border microvilli from the intestinal lumen. **f** Higher magnification of the microvilli highlighting glycocalyx filaments emerging from microvillar distal tips. This panel also shows a vesicle (arrow) and a globular structure (arrowhead) embedded in the glycocalyx network. **g** Image of an unfixed sample confirms a glycocalyx network emerging from microvillar tips. The glycocalyx filaments emerge from the tips of the microvilli and can be distinguished from the lateral links between microvilli (arrowheads). **h** Single 2-nm tomographic slice through the tip of a microvillus showing that the columnar filaments emerge from the membrane (arrow) consistent with a transmembrane mucin. Scale bars: **b, c** = 2 μm; **d** = 1 μm; **e** = 500 nm; **f, h** = 100 nm; **g** = 50 nm.

exclusively from the tip of each microvilli. The density of the filaments made it difficult to determine the exact number of filaments per microvillar tip. The average number of filaments emerging from microvillar tips was $7 \pm 1.5$ ($n = 54$) when counted from freeze-etching cross-fractures that run just above the microvilli and $6.7 \pm 2$ ($n = 22$ microvilli) by counting the number

of filaments that could be visualized emerging from the tips in lateral views of the microvilli. These values are likely to be an underestimate of the number of mucins due to the merging of filaments. Although the glycocalyx network appeared largely devoid of non-filamentous structures, occasionally we observed 20–50 nm granular structures (Fig. 1f, arrowhead) and

50–120 nm vesicles embedded in the glycocalyx matrix (Fig. 1f, arrow), with dimensions of 50–120 nm diameter. The nature and composition of these structures cannot be determined from the freeze-etching images. Some of the structures may correspond to larger exosomes or vesicles that have been reported to originate or shed from microvilli tips[35].

**Glycocalyx filaments make zigzagged lateral contacts**. We next analyzed the lateral interactions between the glycocalyx filaments in 3D space. Filaments emerging from microvillar tips went on to form an interweaving 3D meshwork as seen in both conventional projection views of the freeze-etching replicas (Fig. 2a) as well as in reconstructed full-volume tomograms segmented to visualize glycocalyx network organization (Fig. 2b). The complex zigzagged inter-filament contacts yielded an irregular and complex 3D meshwork. To estimate the mesh or pore size of the network, we measured the maximal distance between filaments outlining randomly selected open spaces within the 3D network (Fig. 2c). The average maximum distance between neighboring filaments was 29 ± 10 nm ($n = 101$; obtained from two montages of the side view of glycocalyx filaments of size $6 \times 1.5$ and $5 \times 1.5$ μm). Qualitatively, it appeared that the inter-filament contacts were more frequent in the middle of the glycocalyx layer than toward the luminal or microvillar ends (Supplementary Fig. 2) and the mesh size in this mid-region was relatively smaller. In tomogram slices and 3D views, we observed filaments converging toward and separating from each other, forming complex cables of different diameters (Fig. 2d–f). In some cases, the lateral contacts between two adjacent filaments showed a narrow gap (Fig. 2e), while in others the separation could not be resolved even by rotating the 3D views of the tomograms (Supplementary Fig. 3).

**Segmentation of the glycocalyx filaments**. The fidelity of freeze-etching replicas is characteristically limited by the granularity and thickness of platinum/carbon film deposited on the frozen biological surface[36–38]. The replica forms a cast-like structure over the surface of the glycoproteins, with replica images showing the platinum/carbon coat rather than the actual structure of the filaments. To partially overcome this intrinsic limitation of the freeze-etching replicas, we segmented the filament impression left on the replica (the void space left after digestion of the tissue) in the reconstructed tomograms (Fig. 2g, blue highlights the filaments and their contacts; gold highlights the replica cast). The views of this segmented void space are more accurate representation of the surface topography and dimensions of the filaments than that observed in conventional views of the replica surface (Fig. 2g). The measured filament diameter, which may represent either individual mucin strands or cables of mucins making lateral contacts, varied from 3 to 15 nm with an average of 5.3 ± 1.3 nm; $n = 51$ (Fig. 2h). Bending of individual filaments appeared to correlate with the deformation or warping of adjacent segments of the network (Fig. 2i), suggesting that filaments and network are under tension. Finally, in close-up views we were able to observe that the filament surface (Fig. 2j, arrowheads) was not homogeneous but rather displayed a quasi-periodic substructure. The quasi-periodic filament substructure is also observed in tomographic reconstructions of the replica (Fig. 2j, arrows).

**Glycocalyx termini form ordered globular tips**. Based on the uniform length of the glycocalyx layer, we next sought to determine its organization at the luminal interface. We found that individual or multiple filaments converged to a globular structure (Fig. 3). Remarkably, all glycocalyx filaments terminated in the same plane, forming a single layer of regularly spaced globular heads much like the surface of a pin brush with ball tips (Fig. 3a–d and Supplementary Fig. 4). The globular tips were relatively uniform, with an average diameter of 10.2 ± 1.2 nm ($n = 82$). We next evaluated the organization of the globular tips and found that their packing varied regionally, ranging from ordered hexagonal packing (Fig. 3e, f, h) to liquid like (Fig. 3g), which we hypothesize is a result of sheer stress on the glycocalyx, to a liquid crystal packing (Fig. 3g). The distance between these glycocalyx filament termini depends on the form of packing, which varied from ordered hexagonal to liquid crystal packing. Using radial distribution function, we estimated that the average distance between termini vary from 31 to 36 nm (Fig. 3e–g). To further evaluate how glycocalyx filament termini are arranged, we performed 3D analysis on a large tomographic volume (Fig. 3i) and obtained a nearest neighbor average spacing between filament termini of 32.0 ± 7.9 nm ($n = 1188$). We also evaluated the distances between termini using an autocorrelation function and radial distribution function on a two-dimensional projection image of the tomographic volume, which showed distances of 38.5 and 40.3 nm, respectively (Supplementary Fig. 5). The difference in the values obtained by these three methods is due to measuring the minimum distance between filaments in the nearest neighbor analysis versus the average distance between filaments in the more inclusive autocorrelation and radial distribution function.

Interestingly, the freeze-etching replicas also show a reduced rate of etching of the ice around the globular tips (Fig. 3b). The higher resistance to water sublimation around the globular tips suggests the existence of some very fine component that is not resolved in the freeze-etching replicas that is limiting water sublimation and keeping the filament termini at a regular distance apart. One possibility is that there is an enrichment of sugar groups decorating the filaments much like a wire pipe brush.

**Probing glycocalyx porosity with fluorescent dextrans**. Finally, to assess the permeability and deformability of the mammalian enteric glycocalyx, we used two approaches. In the first, we removed segments of the small intestine from wild-type mice and fixed them under the same conditions employed for the freeze-etching experiments ("Methods") immediately after euthanasia. These segments were then exposed to dextran solutions of different sizes, conjugated with a fluorophore. We found that a fixable 3 kDa fluorescent dextran, with Stokes radius ≲1.2 nm[39,40], was able to permeate the glycocalyx and decorate the microvilli convoluted surface but not enter the enterocytes (Fig. 4a, b, magenta). Interestingly, the fluorescent signal was stronger at the microvilli than at the glycocalyx layer (Fig. 4b), most likely the result of an increased affinity of the fixable dextran for the reactive groups of the glutaraldehyde fixative present within the tissue or relative exclusion from the highly glycosylated mucin columnar filament network. Conversely, a larger 2000 kDa fluorescent dextran, with an estimated Stokes radius of ~27 nm[39,41] was not able to significantly permeate the glycocalyx, as seen by a clear absence of fluorescence signal (Fig. 4a, b, green). As dextrans are flexible linear molecules, and the actual molecular weights present in a particular sample may have a broad distribution, some fluorescence signal from the 2000 kDa dextran was observed within the glycocalyx layer after a longer exposure to the dextran solution (Fig. 4a, b, green).

**Probing glycocalyx deformability with intravital microscopy**. In our second approach, we sought to investigate the properties of the glycocalyx in a physiological context, in vivo, and developed an approach using intravital confocal microscopy to directly image the intestinal lumen of live, anesthetized mice, after surgical exposure of the intestine[42] and ultimately intestinal lumen.

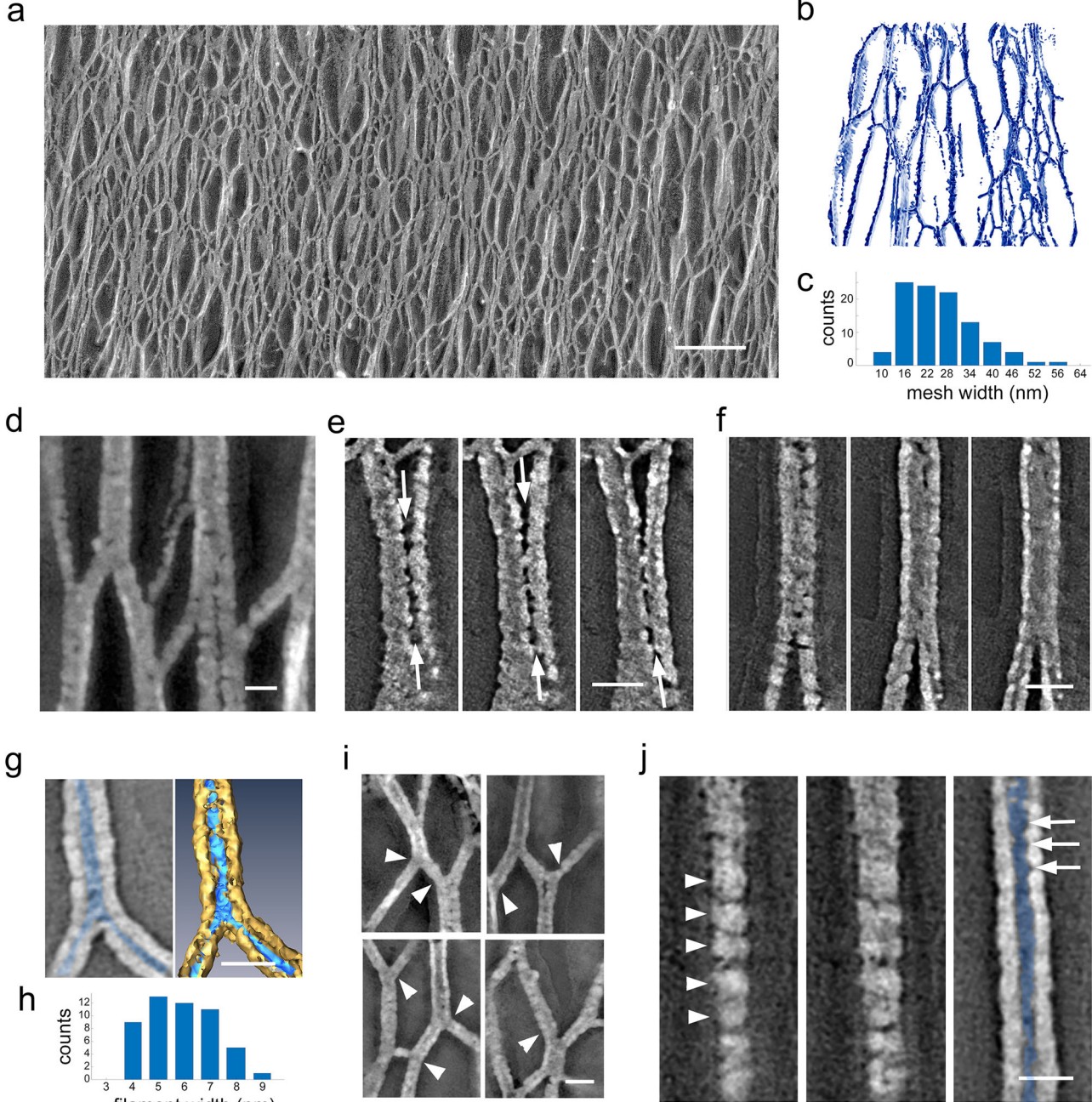

**Fig. 2 The glycocalyx is formed by a three-dimensional network of columnar filaments making lateral contacts. a** Electron micrograph focusing on the complex network of filaments as seen in Fig. 1e that make up the glycocalyx layer. **b** Segmentation of a tomographic volume of the three-dimensional network. **c** Distribution of the pore size between filaments. Mean value = 29 ± 10 nm, $n = 101$. **d–g** Tomograms showing distinct inter-filament interactions. **e** Sequential tomographic slices showing lateral contact (arrows) between adjacent filaments. **f** Sequential tomographic slices showing a contact with no resolvable separation between the filaments. **g** Tomogram slice view of a filament cross-section showing the platinum coating (gray in left panel, gold in right panel) encasing the segmented filament (highlighted in blue). **h** Distribution of filament thickness after accounting for platinum coating. Mean = 5.3 ± 1.3 nm, $n = 51$. **i** Local bending (arrowheads) of individual filaments causes warping of adjacent segments of the network, suggesting that the filaments are under tension. **j** Sequential slices showing a quasi-periodic substructure (arrowheads) on the surface of the replica encasing a filament as well as in the hollow core (blue), which outlines the actual filament topography (arrows). Bar: **a** 200 nm; **d–g**, **i**, **j** 20 nm.

We used transgenic mice expressing mTomato[43], a fluorescently tagged plasma membrane marker, which delineated enterocyte membranes, including the brush border (Fig. 4c). We then directly applied the 3 kDa dextran to the surgically exposed lumen of wild-type mice (Fig. 4d, e). The lumen was rapidly flooded with the dextran, which as before permeated the glycocalyx and filled intermicrovillar spaces (Fig. 4d, e). Importantly, the presence of the fluorescent dextran within the lumen also enabled the visualization erythrocytes by exclusion of fluorescence signal (Fig. 4d, e). During the intravital image acquisitions, the intestine undergoes contractions as well as passive movements caused by breathing and heartbeat, which caused extensive fluid movements around the intestinal villi. Owing to the surgical opening of the intestine, there was often local disruption of blood

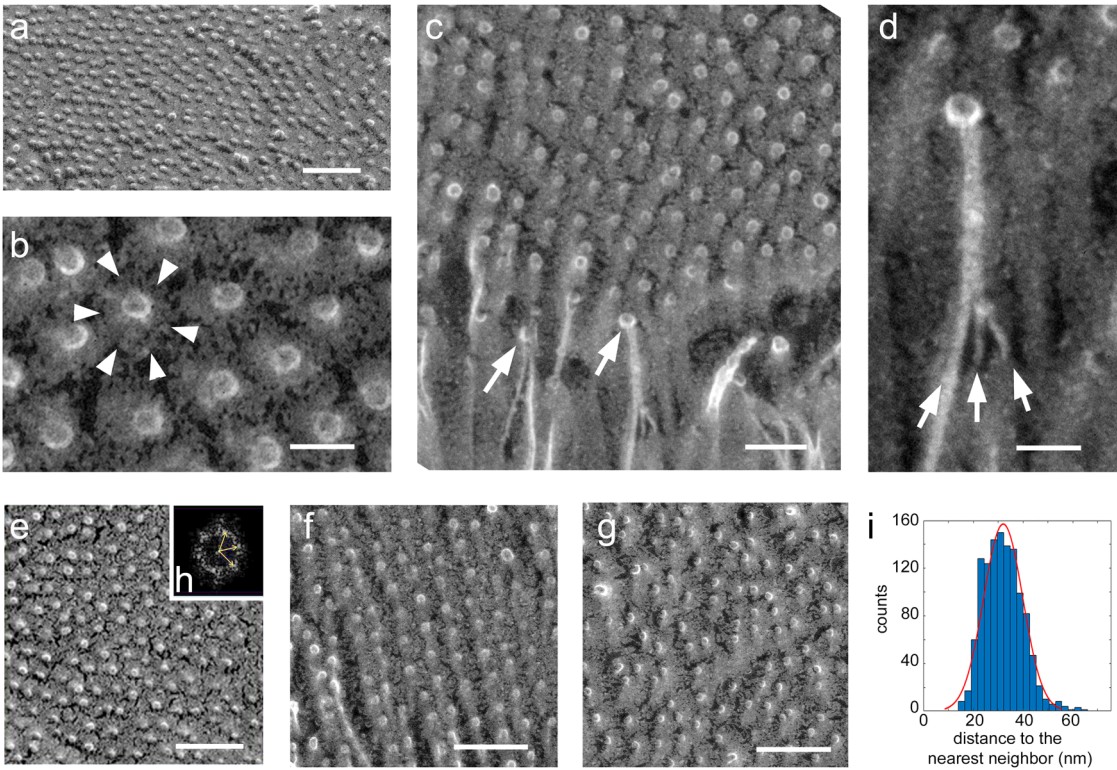

**Fig. 3 Glycocalyx filament termini coalesce to form regularly spaced globular tips. a** Freeze-etching micrographs of the luminal surface of the glycocalyx shows a virtually continuous layer of regularly ordered globular structures, revealing that the glycocalyx filament termini form globular heads that are arranged in a single plane at the luminal interface. **b** Close-up view of the globular structures show that they are surrounded by a radial gradient of platinum deposition (arrowheads) that could indicate that the spaces adjacent to each terminus contain some element that is limiting the etching. **c** An oblique view showing that the single plane of globular structures corresponds to the point where the glycocalyx filament termini abruptly end (arrows). **d** Higher-magnification view showing multiple filaments converging (arrows) before they end into the globular structure. **e–g** The globular structures exhibit distinct local two-dimensional packing arrangements, from hexagonally ordered (**e, f**) to liquid (**g**) packing. Radial distribution function analysis showed termini spacing at 31, 35.4, and 35.5 nm, (**e–g**) respectively. **h** Inset shows the FFT of **e** and indicates hexagonal organization. **i** Distribution, with Gaussian fitting, of nearest neighbor distance between glycocalyx filament termini. Mean = 32.0 ± 7.9 nm, $n$ = 1188. Scale bars: **a** 100 nm; **b** 20 nm; **c** 50 nm; **d** 20 nm; **e**, **f** 100 nm.

vessels and additional fluid flow on the surface of the intestinal epithelium in the field of view. We fortuitously observed that erythrocytes flowing through the lumen, glided along the surface of but never came into contact with the microvillar layer, always maintaining a clear separation, even when squeezed between two villi as in Fig. 4d and Supplementary Movie 1. In instances where erythrocytes packed closely together to the point of contact and mutual deformation, a distinct separation was still maintained from the microvilli, presumably by the glycocalyx. Owing to the diffraction-limited nature of these acquisitions, we could not accurately measure the thickness of the glycocalyx layer. However, the observed gliding and the variations in the separation of the red blood cells from the microvilli surface (Fig. 4d, e, arrows) highlight the flexibility/malleability of the glycocalyx layer and what we interpret as a viscoelastic response in distributing local mechanical stresses throughout the network.

To examine whether the unfixed glycocalyx shows deformations, we performed freeze-etching on directly frozen intestinal epithelium. Segments of the small intestine were gently rinsed and frozen in serum-free culture medium. As expected, during the etching step, the water is sublimated leaving behind the eutectic salt mixture interspersed and decorating the surface of glycocalyx filaments, thus limiting the visualization of the glycocalyx network organization. Nevertheless, we were able to verify that in some regions, in particular where some of the luminal content remained in contact with the glycocalyx layer, the

meshwork appeared compacted and warped (Fig. 4f). We interpret the compaction and warping of the network as structural correlates of the deformability observed in the live imaging.

## Discussion
The key role of the intestinal glycocalyx is to act as size-selective diffusion barrier, excluding particles such as bacteria and viruses and preventing their contact with the enterocyte plasma membrane[26]. While it is known that the glycocalyx is comprised of glycolipids and glycoproteins, including transmembrane mucins, information on the detailed ultrastructural organization and interactions between these components and the resulting architecture of the overall network remains sparse and conflicting. The fine structural features of glycocalyx are not readily distinguishable in conventional transmission electron microscopy because its native structure depends on hydration. In addition, the post-translational addition of large hydrophilic carbohydrates affects protein folding, oligomerization, and aggregation, making it difficult to infer information about the structure and interactions of the glycosylated proteins from their sequence information. Using a combination of freeze-etching and electron tomography, we obtained 3D views of the enteric glycocalyx and details of its nano-scale organization. We also used fluorescent dextrans and intravital imaging[42] to assess the permeability and demonstrate flexibility of the murine enteric glycocalyx.

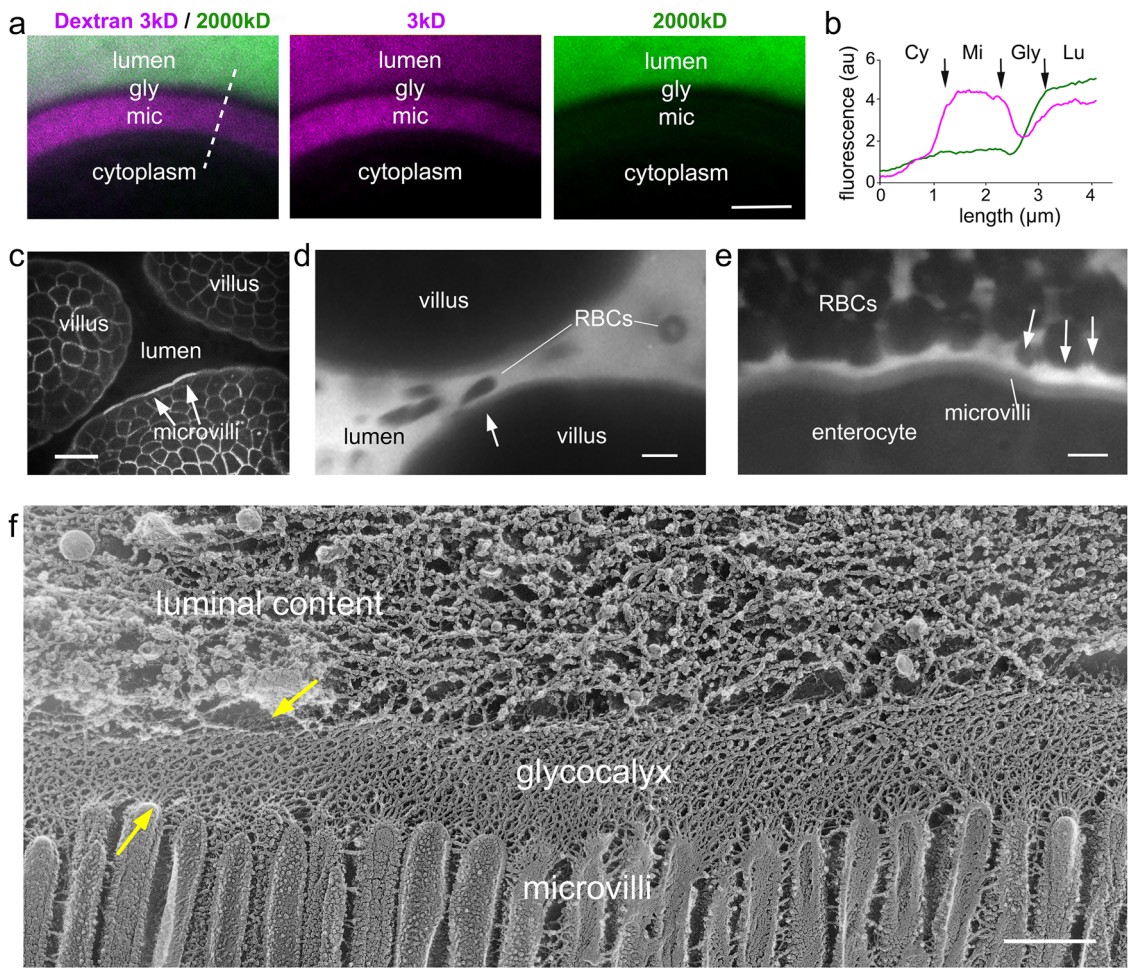

**Fig. 4 Confocal microscopy reveals that the glycocalyx acts as a robust but deformable filter. a** Confocal cross-section across the brush border of fixed intestinal segments exposed to fixable fluorescent dextrans with distinct molecular weights: 3 kDa (magenta) and 2000 kDa (green). The 3 kDa dextran permeates the glycocalyx (gly) layer to accumulate at the microvilli (mic), while the 2000 kDa dextran is mostly excluded from both the glycocalyx layer as well as the microvilli. **b** Fluorescence intensity line scan along the dashed line in **a** showing the variation of dextran fluorescent intensity from cytoplasm (Cy), to microvilli (Mi), to glycocalyx (Gly), and lumen (Lu). **c** Image from intravital microscopy of the intestinal lumen from an mTomato mouse showing the brush border and outlines of the plasma membrane of enterocytes in each villus. **d**, **e** Images from intravital microscopy of the intestinal lumen filled with fluorescent 3 kDa dextran (white). Erythrocytes exclude fluorescent signal. Even when squeezed between adjacent villi (**d**) or tightly packed so they make contact with each other (**e**), erythrocytes are not able to contact the microvillar layer due to the presence of the glycocalyx. Arrows in **d**, **e** point to local variations in the separation of the red blood cells from the microvilli surface consistent with local deformations of the glycocalyx. **f** Freeze-etching replica of unfixed, directly frozen mouse small intestine highlighting that the glycocalyx can be locally deformed (yellow arrows). Scale bars: **a** 2 μm; **c** 10 μm; **d**, **e** 5 μm; **f** 500 nm.

Our data show that the glycocalyx comprises predominantly of columnar filamentous glycoconjugates, exhibiting 3D lateral interactions that result in a densely interlaced structure with pore size of 29 ± 10 nm. The termini of these filamentous proteins appear in the freeze-etching replica as globular structures arranged into a plane lining the intestinal lumen. The average separation between the globular termini was 32.0 ± 7.9 nm. The liquid crystalline packing of the termini likely functions as the first layer of a complex filter. These features are consistent with a molecular sieve that prevents objects with a Stokes radius >30–40 nm from having direct access to the absorptive surface of the enterocytes. Occasionally, we observed ~50–150 nm vesicular structures embedded in the glycocalyx filament meshwork. Some of these structures resemble vesicles reported to shed from microvilli tips[44]. The smaller objects could represent protein aggregates or small exosomes.

The uniform, ~1 μm long, glycocalyx network suggests that the majority of the filaments making up the network are longer

mucins. Because of the extensive lateral interactions, we cannot track individual filaments from the tips of the microvilli to the globular termini. It is likely that shorter mucins such as MUC13[45] also emerge from the tips of microvilli and coalesce with the longer filaments to form the base of the glycocalyx network.

The addition of short glycans along the glycocalyx filaments creates highly hydrophilic regions that help attract and retain water within the glycocalyx layer. This effect is most evident at the apical region of the glycocalyx layer where water appeared to have been retained much more strongly compared to the underlying region in our freeze-etching replicas. Strong sugar–water interaction[46] and structuring of local water molecules[23] are likely major contributors to the observed resistance to water sublimation. Heavy glycosylation also introduces steric hindrance and charge repulsion between component glyco-conjugates[20]. We propose that the alternation of these regions with hydrophilic regions may together result in the distinctive zigzagged interlacing of neighboring glycocalyx filaments to form

the 3D network. We also observed that, as the filaments enter or exit the inter-filament contact segments, they are bent and this bending correlates with deformation and warping of adjacent segments of the network (Fig. 2i). This is suggestive of a network that is under tension whereby the adhesion forces counterbalance the combined forces from bending and pulling of filaments from adjacent filaments of the network. We cannot rule out that fixation before freezing artifactually enhanced or caused some of the inter-filament adherence and bending. Interestingly, glycocalyx filaments on the surface of the intestinal and pancreatic mesothelium that were fixed and processed the same way as the intestine did not show lateral crosslinks. The mesothelium glycocalyx emerged from the plasma membrane of the long filopodia and fanned outwards (Supplementary Fig. 6) without contacting each other or forming a network as observed on the surface of the enterocytes. The non-crosslinked organization of the mesothelial glycocalyx is commensurate with a function focused on lubrication rather than filtration and barrier to pathogens.

Finally, the dynamic and viscoelastic properties of the enteric glycocalyx were observed with intravital imaging. Erythrocytes that were incidentally released into the lumen during the surgical opening of the intestine were seen to glide along the intestinal epithelium without coming into direct contact with the microvilli in the brush border, suggesting that the filaments within the glycocalyx are flexible (Fig. 4). The flow of the erythrocytes across the surface of the glycocalyx layer may also have been assisted by charge repulsion, as the plasma membrane of erythrocytes is also glycosylated[20,47].

Together, freeze-etching, electron tomography, and intravital imaging helped establish a renewed structural understanding of the enteric glycocalyx (Fig. 5) and will provide a concrete basis for future explorations on how the structure of this surface coat correlates with intestinal physiology. The ultrastructural methods utilized in this study may also be applicable in examining glycocalyx structure–function relationship in other epithelia that face similar environmental challenges, such as the airway[48] and the ocular surface[49].

## Methods

**Animals**. All experimental procedures were conducted in accordance with the Guide for the Care and Use of Laboratory Animals by NIH and approved the Animal Care and use Committees for the National Institute on Deafness and Other Communication Disorders (NIDCD ACUC, protocol #1215) and the National Cancer Institute (NCI ACUC, protocol # LCMB-031). mT/mGFP mice were purchased from Jackson Laboratory. For fixed and frozen sample preparation, C57BL/6 mice and rats (Sprague Dawley) of both sexes and age between 1 month and 3 months were euthanized by $CO_2$ asphyxiation and then decapitated. Small intestine was rapidly dissected in phosphate-buffered saline (PBS) or Medium 199 (ThermoFisher) and either directly frozen or fixed for downstream processing. For intravital microscopy, mice were anesthetized by an intraperitoneal (IP) injection of a mixture of ketamine (100 mg kg$^{-1}$) and xylazine (20 mg kg$^{-1}$). The intestine was externalized, and the body temperature of the animals was controlled and maintained at 37–38 °C. At least three animals were imaged per condition tested.

**Fast freezing and freeze-etch electron microscopy**. Samples were fixed with 2% glutaraldehyde and then washed extensively in ddH$_2$O. Samples were fast frozen by rapid contact with the surface of a liquid nitrogen-cooled sapphire block using a Life Cell CF-100 freezing machine. Unfixed intestinal tissues were fine dissected in Medium 199 (ThermoFisher) and were directly frozen as described in the same media or the media was replaced with ddH$_2$O immediately before freezing to reduce the amount of salt frozen with the tissue. Frozen tissues were transferred to a Balzers freeze-fracture apparatus and underwent freeze-fracture at −110 °C followed by freeze-etch at −100 °C for 10 min. Freeze-etched samples were then rotary shadowed with platinum and stabilized with carbon using electron-beam metal-evaporation guns (Cressington Scientific) to create replicas of the exposed surfaces. Samples were subsequently cleaned with sodium hypochlorite and washed with distilled water and then collected onto 300 mesh hexagonal copper grids (Electron Microscopy Sciences). A total of 20 replicas from 6 different mice aged between 1 month and 3 months were produced and examined.

**Data acquisition and electron tomography**. Replicas were examined using a 200 kV JEOL 2100 electron microscope equipped with an Orius 832 CCD camera (Gatan) or a OneView CMOS camera (Gatan). Single images were captured with DigitalMicrograph (Gatan). SerialEM was used to generate montage images and acquire tilt series from −60° to +60° at 1° increments[50]. Montage blending and tomogram reconstruction were done using the IMOD software suite[51]. Total of 13 sets of double-tilt tilt series and 19 montages were acquired, processed, and analyzed.

**Quantitative data analysis**. Fourier analysis of glycocalyx packing was done in DigitalMicrograph (Gatan) or FIJI[52]. Spacing between glycocalyx filament termini of two-dimensional projection images was determined via autocorrelation function[53] (FIJI plugin) and radial distribution function (FIJI plugin). 3D analysis of

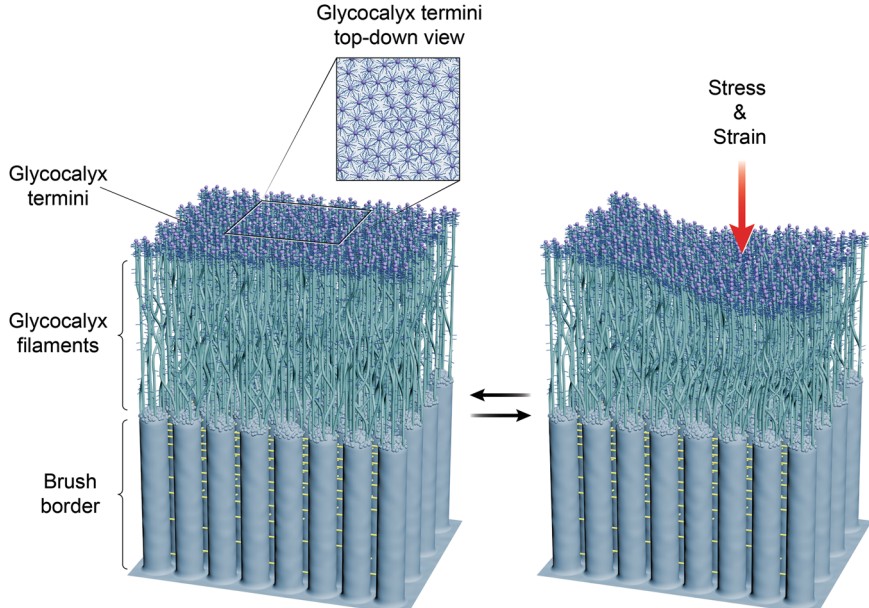

**Fig. 5 A glycocalyx architecture model based on the freeze-etch transmission electron microscopic observations of glycocalyx structures and the dynamics observed by intravital imaging.** Rod-like glycocalyx filaments emerge from microvilli tips above the underlying inter-microvillar cadherin links. These filaments form an intermeshed network through lateral interactions that extends into a flat terminal plane. We argue that the glycocalyx structure network exhibits reversible deformation in response to the local mechanical impact.

distance between filament termini was carried out using a custom MATLAB (Mathworks) script. We manually collected 3D coordinates of individual terminal from a large tomographic volume in FIJI. These coordinates were imported into MATLAB where we use the standard distance between two-point formula for Cartesian coordinate system to calculate and sort distances between two termini to obtain nearest neighbor for each terminal and the average and standard deviation. The MATLAB histfit function was used to produce the histogram that shows the distribution of nearest neighbor distances with a Gaussian fit overlay.

**Segmentation**. Visualization and segmentation of tomograms were performed in Amira (ThermoFisher). 3D renderings of filaments were generated by manual thresholding using the Isosurface module.

**Cryo-sections and immunolabeling**. Intestinal tissues were collected from adult C57BL/6 mice and fixed with 4% paraformaldehyde. After fixation, the samples were put through a sucrose gradient (10% >20% >30%) for cryoprotection. Once the tissue sunk in the 30% sucrose solution, it was placed in a mold in Optimal Cutting Temperature compound (O.C.T) and allowed to freeze on dry ice. Cryo-sections (10–15-μm thick) were then cut and adhered to positively charged glass slides (Electron Microscopy Sciences). Sections were then stored at −80 °C before use. For immunolabeling, sections were first washed with 1× PBS and then permeabilized with 0.5% Triton. For mucin immunostaining, human anti-MUC17 (1:100; Abcam, ab122184) was used to stain for its murine structural homolog MUC3 followed by Alexa-488-conjugated secondary antibody (ThermoFisher). This human anti-MUC17 antibody targets the n-terminal side of the SEA domain[54]. WGA (conjugated with Alexa Fluor 488, catalog #: W11261) was used to label the glycocalyx layer. Samples were counterstained with fluorophore-conjugated phalloidin.

**Intravital microscopy**. Mice were anesthetized by an IP injection of a mixture of ketamine (100 mg kg$^{-1}$) and xylazine (20 mg kg$^{-1}$). The small intestine was surgically externalized, and the epithelium was exposed via a small incision in an area devoid of intestinal content. During the procedure, the epithelial tissue was constantly moistened by applying saline. The anesthetized mouse was placed on the microscopic stage and covered with a heated pad (37 °C) to maintain body temperature. Fixable fluorescent dextran conjugates of 3 or 2000 kDa in size (ThermoFisher D-3305 or D7137) were injected directly into the intestinal lumen via the incision, and the externalized epithelium was then positioned on a coverslip mounted on the stage above the objective and immobilized using custom-made holders. The blood flow was assessed visually by using the eyepiece and only regions close to blood vessels were imaged. The microscope used was a NIKON TiE inverted fluorescence microscope equipped with a Yokogawa CSU-21 spinning disc head and an Andor DU-897 camera. NIKON Elements software was used for image analysis.

**Reporting summary**. Further information on research design is available in the Nature Research Reporting Summary linked to this article.

## Data availability

All data supporting the findings of this study are available from the corresponding author upon reasonable request. Source data for Figs. 2c, h, 3i, and 4b and Supplementary Fig. 5c, d can be found in Supplementary Data 1–6.

## Code availability

The MATLAB script used for nearest neighbor distance calculation is available upon request. Autocorrelation FIJI plugin: http://www.vwalter.fr/ressources/software/acf_imagej/. Radial distribution function FIJI plugin: https://imagejdocu.tudor.lu/doku.php?id=macro:radial_distribution_function.

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

## Acknowledgements

This research was supported by the NIH, NIDCD Intramural Research Program (Z01 DC 000002), the NCI Center for Cancer Research Intramural Research Program (ZIA BC 011682), the NIDCD Advanced Imaging Core (ZIC DC000081), and the São Paulo Research Foundation (FAPESP grant 13/22816-2). We thank Ethan Tyler for the diagram drawings, Manoj Yadav for help with the dextran solutions, Bryan Millis for preliminary work on intestinal epithelium in the laboratory, and Ronald Petralia for critical reading of the manuscript.

## Author contributions

W.W.S., E.S.K., R.C. and B.K. designed experiments and prepared samples. Image acquisition, processing, and analysis were carried out by W.W.S., E.S.K. and B.K. Image segmentation was done by A.L.-M. Intravital imaging and analysis were designed and performed by S.E., R.W. and B.K. Experiments and imaging of mesothelial cells were performed by A.S. and B.K. W.W.S., E.S.K., S.E. and B.K. wrote the manuscript and all authors participated in the final editing.

## Competing interests

The authors declare no competing interests.
