## [Peer Review File · Communications Biology]

Reviewers' comments:

Reviewer #1 (Remarks to the Author):

In the intestine, the glycocalyx consist of glycolipids and glycoproteins that form a barrier at the interface between the microvilli-rich apical domain of epithelial cells and the luminal content such as gut bacteria. Due to the complex biochemistry of membrane mucins that constitute the glycocalyx, there is limited knowledge about the structure, organization and function of the glycocalyx. Sun et al used freeze-fracturing and freeze-etching combined with electron tomography to characterize the nanoarchitecture of the enteric glycocalyx of the mouse. The authors also used intravital imaging to investigate the barrier function of the glycocalyx.

The authors conclude that the glycocalyx functions as a deformable size-exclusion filter for luminal contents. While the concept of the glycocalyx as a diffusion barrier is not new (the authors refer to Frey et al. *J Exp Med.* 1996), this manuscript provides quantitative data that proves this concept to be true. The conclusions presented in this manuscript are mostly well supported in the presented qualitative and quantitative data. The study provides relevant and timely insight into the organization of the intestinal glycocalyx, a topic that has long been neglected. In particular, the authors present important structural information that will help shed light on the function of glycocalyx on epithelial surfaces.

However, a number of technical and conceptual issues need to addressed. The authors should try to investigate or at least discuss the relationship between different membrane mucins expressed in intestinal glycocalyx. Moreover, conclusions are often based solely on observations in electron micrographs. While electron microscopy provides detailed structural information, observations should be followed up by additional experiments using other methods. Finally, the authors claim that the glycocalyx is a size-exclusion filter for luminal contents but fail to devise an appropriate biologically relevant experimental setup to test this hypothesis.

In summary, this is a well-written manuscript that after minor revision addressing the concerns raised below, will be an important contribution to the field of cell biology and will also prove useful for researchers interested in investigating host-microbe interactions involving membrane mucins expressed on epithelial cells.

Major issues:

1. The gene encoding human MUC17 lies within the locus 7q22. In the same genomic locus, upstream of MUC17, two additional genes encoding protein products with typical proline, serine and threonine (PTS)-rich domains and transmembrane domains, can be found. These two genes encode MUC3 and MUC12 that resemble MUC17 in terms of domain organization and sequence identity. For example, MUC3, MUC12 and MUC17 all harbor extracellular SEA domains and intracellular PDZ domains (Malmberg et al., *Biochem J.* 2008 and Pelaseyed et al., *Am J Physiol Cell Physiol.* 2013). In the mouse, the gene designated *Muc3* is in fact murine *Muc17*, based on location in the chromosome (two genes encoding PTS-rich transmembrane proteins are located upstream) and sequence analysis of the intracellular domain. The authors are in fact aiming to stain for murine *Muc17* in sections from mouse small intestine. This should be clarified.

2. In fig 1b, the glycocalyx in mouse small intestine is stained for *Muc17* and the brush border membrane is visualized using the lectin WGA. There are a number of issues that should be addressed, either experimentally or in the Discussion section, in order to provide a better understanding of the organization of the apical glycocalyx.

2a) Micrographs show long filaments of membrane mucin that can be distinguished from tip link

complexes of microvilli. A closer examination also reveals additional filamentous structures below the tip links, in fact along the length of microvilli (for an example see fig 1d). Human MUC17, or murine Muc17, is one major glycoprotein component of the glycocalyx. However, the authors do not investigate or discuss other intestinal membrane mucins, such as MUC3 and MUC12 in human and/or Muc13 in mouse. As mentioned, MUC3 and MUC12 are genetically and structurally related to MUC17, while MUC13 is a short membrane mucin of ca 500 amino acids. It would be important to address the expression and distribution of at least some of these other membrane mucins within the glycocalyx using available antibodies and to discuss if some or any of these membrane mucins account for additional features observed in electron micrographs.

2b) The polyclonal MUC17 antibody (Abcam, ab122184) used in the study is not clearly defined and more importantly, has not been validated using common antibody validation steps. Because a Muc17 knock-out mouse is not yet available to the community, peptide block or at least stainings showing tissue as well as cell specificity should be added to the manuscript. Antibody validation is important in order to avoid misinterpretations due to cross-reactivity with other SEA-type membrane mucins. The epitope (residues 4122-4262) recognized by the antibody is within the N-terminal portion of the SEA domain, a domain that is structurally conserved amongst SEA-type membrane mucins (see Macao et al., *Nat Struct Mol Biol.* 2006 and Johansson et al., *J Mol Biol.* 2008).

3. The authors mention speculate about vesicles and globular structures embedded in the glycocalyx network (fig 1e) but do not provide any experimental evidence for this claim. Regarding the smaller globular structures (20-30 nm), the authors should make an attempt to discuss the identity of these globular structures. Are there any globular structures in membrane mucins that could account for this observation? Could the observed globular structures be dimers or even oligomers of neighboring extracellular SEA domains?

4. The section "Bending of individual filaments and warping of the network." is very brief and fails to convince the reader of a correlation between bending of filaments and deformation of the membrane mucin network. The authors should expand on this issue and provide convincing quantitative data that actually shows this correlation. For example, can the authors quantify more bending of filaments when dense luminal content exerts pressure on the glycocalyx as in Fig 4f?

5. The authors present the hypothesis that membrane mucins in the glycocalyx act as barriers that protect the microvilli. To test this hypothesis, they use intravital microscopy and claim that erythrocytes are excluded from the 3kDa dextran-positive zone and are thus unable to make contact with the underlying microvilli. Figure 4d is not very convincing and while 4e does show a separation between erythrocytes and the brush border membrane, the relevance assessing barrier function using erythrocytes is questionable. The authors should set up a biologically relevant model to test the barrier function of the glycocalyx, for example by using bacteria that colonize the mouse small intestine.

Minor issues:

1. Figure 1b would be much more informative if the upper panel was a high-quality image (compared to lower panel, upper panel seems to be of lesser quality), and if single channels were visualized. Alternatively, single-channel images could be moved to Supplementary materials section.

2. Under "Material and Methods" > "Quantitative data analysis", the authors mention a custom MATLAB script to calculate nearest neighbor distance. The authors should provide details about this script.

3. In figure text, figure 1j reads 1i. Change to 1j.

Reviewer #2 (Remarks to the Author):

This is an excellent piece of work using a technique that has gone out of fashion. Sugars wash out of samples in resin EM embedding, here that is negated, hydrated cryo-TEM is not ready yet, and won't be for full biological studies for a good while. The analysis performed is trying to answer the pertinent questions. These sections need at least clarity (in the methods), and perhaps further analysis of the same data.

Introduction

Please reference the technique used here

Fig 1

Please add the 'brush boarder label to a/c

Fig 2a is overlapping with fig 1d's left hand side, please state this.

Results:

Lack of vesicles, could this be washing out?

Please also state number of tomograms/images/animals analysed not just pores/vilie etc.

Data analysis

Nearest neighbour spacing:

I took figure 1, box an area then performed the analysis that Squire's group used (2001 and 2011; FFT and Autocorrelation). The result is below. This gives a clear regular spacing at 90nm (and probably some others). This type of analysis would be possible on the different layers (and should pull out the differential spacing in the middle. AC is easier to interpret in heterogeneous samples compared to FFTs. The nearest neighbour that you have used needs more explanation: clarify if it was on the 2D or 3D datasets, or 2D slices from the 3D datasets. With the plan view of the tips: please state the mean spacing even in the non-structured version (e.g. fig 3g), perhaps via a radial distribution function.

'Fortuitously orientated samples', please speculate caveat etc in discussion on these.

Stokes radius <2nm : This is known for 3kDa dex (numerous references) and is much smaller than <<2, probably nearer 1.2nm.

Pore size by exclusion? Is it possible to do the following? You know the concentration, you know the size (ish) therefore the pore size (partition coefficient) can be worked out as it has the same concentration as the lumen and the microvilli.

Discussion:

"Other approaches such as cryo-electron microscopy, which require in vitro protein preparations..." This is not actually true, as in-situ cryo EM could be theoretically be used, but the point is very valid! Nobody has observed glyx with hydrated cryo-EM and it would need things like phase plates and lamella milling to have a chance. For this sub fibre sub tomogram averaging would need to be done (to overcome the heterogeneity) and even then in-situ cryoEM is in its infancy.

Please spend time discussing: the non-fixed samples vs fixed samples, any changes in structure you might have from ionic concentration/sample prep.

Referee expertise:

Referee #1: Mucins, mucus layer, intestinal epithelial cells

Referee #2: 3D reconstruction with electron tomography

Reviewers' comments:

Reviewer #1 (Remarks to the Author):

In the intestine, the glycocalyx consist of glycolipids and glycoproteins that form a barrier at the interface between the microvilli-rich apical domain of epithelial cells and the luminal content such as gut bacteria. Due to the complex biochemistry of membrane mucins that constitute the glycocalyx, there is limited knowledge about the structure, organization and function of the glycocalyx. Sun et al used freeze-fracturing and freeze-etching combined with electron tomography to characterize the nanoarchitecture of the enteric glycocalyx of the mouse. The authors also used intravital imaging to investigate the barrier function of the glycocalyx.

The authors conclude that the glycocalyx functions as a deformable size-exclusion filter for luminal contents. While the concept of the glycocalyx as a diffusion barrier is not new (the authors refer to Frey et al. *J Exp Med.* 1996), this manuscript provides quantitative data that proves this concept to be true. The conclusions presented in this manuscript are mostly well supported in the presented qualitative and quantitative data. The study provides relevant and timely insight into the organization of the intestinal glycocalyx, a topic that has long been neglected. In particular, the authors present important structural information that will help shed light on the function of glycocalyx on epithelial surfaces.

However, a number of technical and conceptual issues need to be addressed. The authors should try to investigate or at least discuss the relationship between different membrane mucins expressed in intestinal glycocalyx. Moreover, conclusions are often based solely on observations in electron micrographs. While electron microscopy provides detailed structural information, observations should be followed up by additional experiments using other methods. Finally, the authors claim that the glycocalyx is a size-exclusion filter for luminal contents but fail to devise an appropriate biologically relevant experimental setup to test this hypothesis.

In summary, this is a well-written manuscript that after minor revision addressing the concerns raised below, will be an important contribution to the field of cell biology and will also prove useful for researchers interested in investigating host-microbe interactions involving membrane mucins expressed on epithelial cells.

Major issues:

1. The gene encoding human MUC17 lies within the locus 7q22. In the same genomic locus, upstream of MUC17, two additional genes encoding protein products with typical proline, serine and threonine (PTS)-rich domains and transmembrane domains, can be found. These two genes encode MUC3 and MUC12 that resemble MUC17 in terms of domain organization and sequence identity. For example, MUC3, MUC12 and MUC17 all harbor extracellular SEA domains and intracellular PDZ domains (Malmberg et al., *Biochem J.* 2008 and Pelaseyed et al., *Am J Physiol Cell Physiol.* 2013). In the mouse, the gene designated *Muc3* is in fact murine *Muc17*, based on location in the chromosome (two genes encoding PTS-rich transmembrane proteins are located upstream) and sequence analysis of the intracellular

domain. The authors are in fact aiming to stain for murine Muc17 in sections from mouse small intestine. This should be clarified.

Following the reviewer's suggestion, we added a comment and references indicating that Muc3 is in fact murine Muc17, and because our antibody is against the SEA domain common to other mucins, we are likely recognizing additional mucins. This further validates the point we tried to make with this labeling, which is to show that our dissection and tissue preparation preserves the membrane anchored mucin layer.

2. In fig 1b, the glycocalyx in mouse small intestine is stained for Muc17 and the brush border membrane is visualized using the lectin WGA. There are a number of issues that should be addressed, either experimentally or in the Discussion section, in order to provide a better understanding of the organization of the apical glycocalyx.

2a) Micrographs show long filaments of membrane mucin that can be distinguished from tip link complexes of microvilli. A closer examination also reveals additional filamentous structures below the tip links, in fact along the length of microvilli (for an example see fig 1d). Human MUC17, or murine Muc17, is one major glycoprotein component of the glycocalyx. However, the authors do not investigate or discuss other intestinal membrane mucins, such as MUC3 and MUC12 in human and/or Muc13 in mouse. As mentioned, MUC3 and MUC12 are genetically and structurally related to MUC17, while MUC13 is a short membrane mucin of ca 500 amino acids. It would be important to address the expression and distribution of at least some of these other membrane mucins within the glycocalyx using available antibodies and to discuss if some or any of these membrane mucins account for additional features observed in electron micrographs.

Unfortunately, we are unable to distinguish different SEA domain containing transmembrane mucins with our approach. In addition, we have not found commercially available antibodies for each mouse mucin. We have added a sentence to our discussion indicating that we cannot exclude that some of the filaments emerging from the microvilli tip or lateral surface are short mucins.

2b) The polyclonal MUC17 antibody (Abcam, ab122184) used in the study is not clearly defined and more importantly, has not been validated using common antibody validation steps. Because a Muc17 knock-out mouse is not yet available to the community, peptide block or at least stainings showing tissue as well as cell specificity should be added to the manuscript. Antibody validation is important in order to avoid misinterpretations due to cross-reactivity with other SEA-type membrane mucins. The epitope (residues 4122-4262) recognized by the antibody is within the N-terminal portion of the SEA domain, a domain that is structurally conserved amongst SEA-type membrane mucins (see Macao et al., Nat Struct Mol Biol. 2006 and Johansson et al., J Mol Biol. 2008).

The epitope of this commercially available anti-human MUC17 antibody contains the MUC17-S1 epitope listed in Pelaseyed et al., Am J Physiol Cell Physiol. 2013, which is located at the n-terminal side of the SEA domain. Due to the prevalence of the SEA domain amongst members of transmembrane mucins (including MUC3, 13, 12 & 17), it is very likely that this antibody also binds to SEA domain of other transmembrane mucins as mentioned in our response to comment #1.

3. The authors mention speculate about vesicles and globular structures embedded in the glycocalyx network (fig 1e) but do not provide any experimental evidence for this claim. Regarding the smaller globular structures (20-30 nm), the authors should make an attempt to discuss the identity of these globular structures. Are there any globular structures in membrane mucins that could account for this

observation? Could the observed globular structures be dimers or even oligomers of neighboring extracellular SEA domains?

We appreciate the interesting possibility raised by the reviewer. The globular structures we describe are rare, and more importantly we do not have any means to determine their molecular composition or nature. We added a comment about this limitation in the discussion.

4. The section “Bending of individual filaments and warping of the network.” is very brief and fails to convince the reader of a correlation between bending of filaments and deformation of the membrane mucin network. The authors should expand on this issue and provide convincing quantitative data that actually shows this correlation. For example, can the authors quantify more bending of filaments when dense luminal content exerts pressure on the glycocalyx as in Fig 4f?

We recognize that the bending and warping of the glycocalyx framework are descriptive interpretation of our images. We also do not have means to quantify pressure distribution. Such measurements would require additional experimental procedures and maybe instrumentation like AFM. We feel such experiments are beyond the scope of this first structural study.

5. The authors present the hypothesis that membrane mucins in the glycocalyx act as barriers that protect the microvilli. To test this hypothesis, they use intravital microscopy and claim that erythrocytes are excluded from the 3kDa dextran-positive zone and are thus unable to make contact with the underlying microvilli. Figure 4d is not very convincing and while 4e does show a separation between erythrocytes and the brush border membrane, the relevance assessing barrier function using erythrocytes is questionable. The authors should set up a biologically relevant model to test the barrier function of the glycocalyx, for example by using bacteria that colonize the mouse small intestine.

Indeed, it would be desirable to use either biophysically better-defined probes or biologically relevant bacteria. Again, our laboratory is not equipped to perform these experiments.

Minor issues:

1. Figure 1b would be much more informative if the upper panel was a high-quality image (compared to lower panel, upper panel seems to be of lesser quality), and if single channels were visualized.

Alternatively, single-channel images could be moved to Supplementary materials section.

We made the two panels to have equal magnification and directly comparable.

2. Under “Material and Methods” > “Quantitative data analysis”, the authors mention a custom MATLAB script to calculate nearest neighbor distance. The authors should provide details about this script.

As requested, a description of the MATLAB script was added to the Material and Methods section.

3. In figure text, figure 1j reads 1i. Change to 1j. *figure 2j needs to be changed to figure 2j*

We corrected our oversight.

Reviewer #2 (Remarks to the Author):

This is an excellent piece of work using a technique that has gone out of fashion. Sugars wash out of samples in resin EM embedding, here that is negated, hydrated cryo-TEM is not ready yet, and won't be for full biological studies for a good while. The analysis performed is trying to answer the pertinent questions. These sections need at least clarity (in the methods), and perhaps further analysis of the same data.

Introduction

Please reference the technique used here

As requested, we added a brief description of how freeze-etch works to the introduction.

Fig 1

Please add the 'brush boarder label to a/c

The small size of the panels did not allow us to add another label. Instead we changed the description of the legend to indicate that the microvilli collectively represent the brush-border.

Fig 2a is overlapping with fig 1d's left hand side, please state this.

We have now stated that in Figure 2 legend that panel a is a high-mag, close-up view of a region in Figure 1d.

Results:

Lack of vesicles, could this be washing out?

Please also state number of tomograms/images/animals analysed not just pores/vilie etc.

The number of animals, replicas, tomograms, and images was added to the methods section.

Data analysis

Nearest neighbour spacing:

I took figure 1, box an area then performed the analysis that Squire's group used (2001 and 2011; FFT and Autocorrelation). The result is below. This gives a clear regular spacing at 90nm (and probably some others). This type of analysis would be possible on the different layers (and should pull out the differential spacing in the middle. AC is easier to interpret in heterogeneous samples compared to FFTs. The nearest neighbour that you have used needs more explanation: clarify if it was on the 2D or 3D datasets, or 2D slices from the 3D datasets. With the plan view of the tips: please state the mean spacing even in the non-structured version (e.g. fig 3g), perhaps via a radial distribution function.

As requested, we performed autocorrelation function and radial distribution function on our data. The results are now included in the text and in supplementary figure 5.

'Fortuitously orientated samples', please speculate caveat etc in discussion on these.

We removed the term "fortuitously" and rewrote the sentence to avoid any ambiguity.

Stokes radius <2nm : This is known for 3kDa dex (numerous references) and is much smaller than <<2, probably nearer 1.2nm.

We have added references to and edited the manuscript to indicate 3kDa dextran is ≈ 1.2 nm.

According to ThermoFisher dextran product sheet, the 3kDa dextran is actually a mixture of 1.5kDa to 3kDa dextran. Hence, we stated in the manuscript that the approximate Stokes radius for 3kDa dextran is less-than or approximately 1.2 nm.

Pore size by exclusion? Is it possible to do the following? You know the concentration, you know the size (ish) therefore the pore size (partition coefficient) can be worked out as it has the same concentration as the lumen and the microvilli.

Unfortunately, we are unable to measure with precision the concentration and determine partition coefficient.

Discussion:

"Other approaches such as cryo-electron microscopy, which require in vitro protein preparations..."

This is not actually true, as in-situ cryo EM could be theoretically be used, but the point is very valid!

Nobody has observed glyx with hydrated cryo-EM and it would need things like phase plates and lamella milling to have a chance. For this sub fibre sub tomogram averaging would need to be done (to overcome

the heterogeneity) and even then in-situ cryoEM is in its infancy.

This is an excellent point, it is likely possible to examine in the future the mucins using a FIB-SEM lamella, phase-plate based cryo-tomography approach. We have adjusted the discussion to better state the current advantages of freeze-etching and removed comments on cryo-electron microscopy for simplicity.

Please spend time discussing: the non-fixed samples vs fixed samples, any changes in structure you might have from ionic concentration/sample prep.

We have updated the results to directly compare the fixed vs unfixed sample and added a discussion on the cause of the difference in the glycocalyx filament network including fixation artifacts, the presence of the luminal contents, or the pure water environment.

REVIEWERS' COMMENTS:

Reviewer #1 (Remarks to the Author):

The authors conclude that the glycocalyx functions as a deformable size-exclusion filter for luminal contents. While the concept of the glycocalyx as a diffusion barrier is not new (the authors refer to Frey et al. J Exp Med. 1996), this manuscript provides quantitative data that proves this concept to be true. The conclusions presented in this manuscript are mostly well supported in the presented qualitative and quantitative data. The study provides relevant and timely insight into the organization of the intestinal glycocalyx, a topic that has long been neglected. In particular, the authors present important structural information that will help shed light on the function of glycocalyx on epithelial surfaces.

A number of technical and conceptual issues were raised. The authors were urged to investigate and/or discuss the relationship between different membrane mucins expressed in intestinal glycocalyx. Moreover, as many conclusions were based solely on observations in electron micrographs, the authors were urged to perform additional experiments to support their conclusions. The authors have clearly discussed some of the technical limitations of studying the glycocalyx. After evaluating the revised manuscript and the authors' rebuttal letter, I believe that this work contributes with new knowledge regarding the ultrastructure of the glycocalyx and membrane mucins that constitute this structure.

Reviewer #2 (Remarks to the Author):

The Authors have satisfactorily addressed my concerns.

--

"The calculated average for the cross-sectional pore or mesh size for the network was 29 ± 10 nm"

Please can you confirm/clarify if this is the mean from the samples of the nearest neighbour, or the mean of the neighbours... the first is not the same as the mean pore size which is implied. This would then explain the difference in the RDF and ACF quite nicely.